# GRADIENT-FREE ADVERSARIAL ATTACK ON TIME SERIES REGRESSION: TARGETING XAI EXPLANATIONS

## ABSTRACT

Explainable Artificial Intelligence (XAI) sheds light on the decision-making ground of black-box models by offering explanations. These explanations need to be robust for trustworthy time series regression applications in high-stake areas like medicine or finance, which yet remains largely unexplored. Furthermore, most adversarial attack methods currently rely on white-box strategies, which require access to gradient information from both the model and the XAI method. In real-world scenarios, such information is often difficult or impossible to obtain. To address these challenges, we propose a novel gradient-free adversarial attack method specifically designed for time series explanations, targeting non-differentiable XAI techniques. To enhance the effectiveness of our method for time series data, we introduce an attack objective function based on Dynamic Time Warping (DTW). Additionally, we implement an explanation-based local attack strategy, which ensures that the adversarial perturbations remain imperceptible within the time series data. In our experiments, we generate adversarial examples to attack four different XAI methods across three black-box models, using two time series datasets. The results reveal the vulnerability of current non-differentiable XAI methods. Furthermore, by comparing our approach with existing attack methods, we demonstrate the superiority of our proposed objective function and local attack strategy.

## 1 INTRODUCTION

The "black box" nature of Artificial Intelligence (AI) has raised public concerns about its lacking of transparency and explainability in decision-making process. To address this issue, Explainable AI (XAI) has emerged, which aims to provide clear and easy-to-understand explanations of AI's decision making in order to eliminate the public's concerns about AI, enhance users' trust, and further promote the application of AI in critical fields such as military, healthcare, and finance (Ali et al., 2023). Many XAI methods are developed and applied nowadays, such as attribution-based methods or example-based methods, which, however, generally lack of the robustness of the explanation. Taking the medical field as an example, doctors can diagnose or make decisions based on XAI-generated explanations, and if a small perturbation (e.g., adversarial attack) is made to the input sample can generate incorrect explanations. Making decisions based on such compromised explanations could have serious consequences, potentially endangering a patient's life (van der Velden et al., 2022). This vulnerability raises questions about the validity and reliability of XAI-generated explanations when faced with adversarial attacks, emphasizing the need for more robust XAI methods.

Major research on XAI robustness have focused on adversarial attacks (including adversarial example generation, adversarial model generation, or data poisoning) and the corresponding defense methods (Baniecki and Biecek, 2024). The most widely studied attack approach in this area is adversarial example generation, where the core idea is to introduce perturbations to the input data that alter the model's explanations while leaving its classification results unchanged.(Ghorbani et al., 2019). Corresponding defense methods seek to make the explanations as stable as possible under such situations. However, most methods for generating adversarial examples rely on gradient descent, which is a white-box attack technique that requires access to the model's loss function and internal gradient information of both the model and the XAI method(Huang et al., 2023). This requirement for prior knowledge of gradients is often difficult to fulfill in practical applications (Akhtar and Mian, 2018).

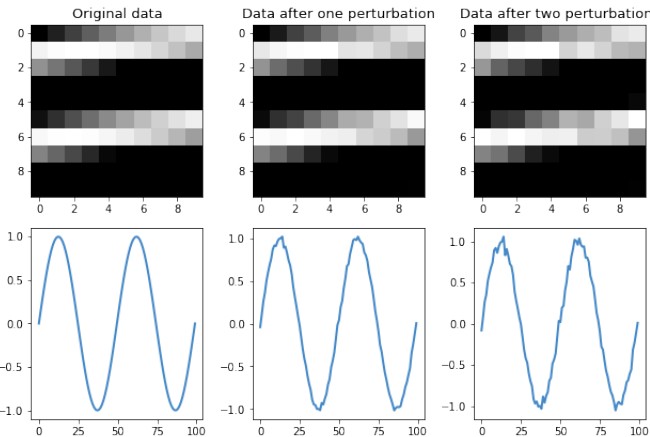

Figure 1: This figure shows the same data after a single perturbation and a double perturbation when they are in image form and line form respectively.

To address these issues, we propose a gradient-free algorithm based on Particle Swarm Optimization (PSO), which searches for adversarial examples without needing gradient information, making it more effective in real-world scenarios and with non-differentiable XAI techniques.

On the other hand, most current research on the robustness of XAI has been concentrated in the image domain, with very limited attention given to time series data. Time series data present a unique challenge for adversarial attacks due to the way perturbations are perceived by humans. In image data, slight changes in pixel colors may go unnoticed by the human eye, but humans can easily detect subtle perturbations in the patterns or lines of a time series. For instance, as shown in Fig.**??**, while small variations in image pixel colors are hard to perceive, humans are more sensitive to perturbations in time series lines (Yang et al., 2022). If the common strategy of perturbing the entire sample (which is widely applied in current XAI attacks) is used for time series data, the generated adversarial perturbations are more likely to be detected, reducing the effectiveness of the attack. Therefore, we further design a local attack strategy based on explanation, where only some time points in the time series are selected to be perturbed according to the results of XAI. This strategy not only minimizes the magnitude of perturbations, making them harder to detect, but also increases the efficiency of the attack. Additionally, another challenge arises from the fact that existing objective functions used in adversarial attacks are often unsuitable for time series data. To tackle this, we leverage Dynamic Time Warping (DTW) to design a more appropriate objective function, ensuring that the attack is more effective for time series data.

In summary, the research goal of this paper is to design a new attack method against XAI on time series regression problems, demonstrating the vulnerability of XAI methods. The contributions of this paper are summarized as follows:

- **We propose a gradient-free adversarial attack method to test the robustness of XAI explanations.** By introducing the PSO search method, we can attack non-differentiable XAI methods without requiring gradient computation. Additionally, we design a DTW-based objective function and a local attack strategy aimed at enhancing the effectiveness of the attack in time series regression problems.

- **Our experimental study demonstrates the effectiveness of our proposed attack.** The proposed method successfully attacks non-differentiable XAI explanations, validating its capability to test XAI robustness. Furthermore, the explanation-based local attack strategy not only reduces computational overhead but also makes the attack more stealthy while achieving results comparable to those of global attack strategies.

The remainder of this paper is organized as follows. Sec.2 introduces some of the work related to this paper. And we will briefly introduce the background and concepts in Sec.3 Sec.4 describes our proposed attack method, and the experiment results in Sec.5. Finally, the conclusion is drawn in Sec.6.

## 2 RELATED WORK

Adversarial attacks on XAI explanations can be categorized into three main approaches: generating adversarial examples, generating adversarial models, and data poisoning (Baniecki and Biecek, 2024). We focus on adversarial example generation because it was first proposed as a method to demonstrate the fragility of XAI explanations (Ghorbani et al., 2019), and it remains the most extensively researched form of adversarial attack. The work designing an iterative attack based on gradient descent against simple gradient, integrated gradient and deeplift, with analysis of Hessian matrix to reveal robustness is a common challenge in current XAI methods (Ghorbani et al., 2019). These attacks are effective because the nonlinear and non-smooth properties of neural networks cause the gradient to vary significantly over small input perturbations (Ghorbani et al., 2019; Dombrowski et al., 2019; Wang et al., 2020). In response, various defense strategies have been developed to counter these attacks, including smoothing activation functions (Dombrowski et al., 2019), adding regularization terms (Wang et al., 2020; Joo et al., 2023), applying weight decay (Dombrowski et al., 2022), and minimizing the network's Hessian matrix (Dombrowski et al., 2022).

However, a significant limitation of gradient-based attacks is the requirement for knowledge of the model's loss function and gradient, which is often inaccessible in practical applications (Akhtar and Mian, 2018). Moreover, when the XAI method is non-differentiable (as is the case with perturbation-based methods), these gradient-based attacks become ineffective because the explanation's gradient with respect to the input cannot be calculated. In such cases, researchers have resorted to adversarial model generation (Slack et al., 2020) or data poisoning attacks (Baniecki et al., 2022). Recently, adversarial example generation methods for non-differentiable XAI have emerged, utilizing gradient-free techniques such as Genetic Algorithms (GA) to overcome the limitations of traditional approaches (Huang et al., 2023; Baniecki and Biecek, 2022). However, these gradient-free methods have so far been applied to image and tabular data (Huang et al., 2023; Baniecki and Biecek, 2022), with no work addressing time series data. Therefore, our objective is to propose a gradient-free adversarial example generation method specifically tailored for time series XAI tasks.

## 3 PRELIMINARIES

### 3.1 ATTRIBUTION BASED XAI

There has been extensive research on obtaining explanations from black-box models, with attribution-based methods being the focus of this paper. Attribution-based methods aim to identify which features have the greatest impact on a model's decisions and attempt to quantify the contribution of each feature to the model's predictions. These methods can be divided into two main categories: gradient-based methods and perturbation-based methods. Gradient-based methods assess the importance of features by using the model's gradient with respect to input features. In contrast, perturbation-based methods determine feature importance by masking or modifying features and observing the resulting changes in the model's predictions. While gradient-based methods are computationally more efficient, their results are often not directly aligned with changes in the model's output (Ancona et al., 2018). Furthermore, gradient-based methods are only applicable to differentiable models, limiting their versatility compared to perturbation-based methods.

In this paper, we propose adversarial attacks on perturbation-based XAI methods, with a focus on two widely-used techniques: Locally Interpretable Model-Agnostic Explainer (LIME) and SHapley Additive exPlanation (SHAP). Additionally, to compare the effectiveness of our attacks on gradient-based methods, we include saliency maps and SmoothGrad in the analysis.

- **Saliency Map (SM)** applies a first-order linear approximation of the model to detect the sensitivity of the score to perturbations in each input dimension. Given input $X$, its $i$-th feature's importance score $I(x_i)$ can be computed as the gradient of the model output $f(X)$ with respect to $x_i$ (Baehrens et al., 2010):

$$I(x_i) = \frac{\partial f(X)}{\partial x_i} \tag{1}$$

- **SmoothGrad (SG)** is an improvement on SM by adding random noise around the input features for $n$ times and then averaging the gradient of the noisy samples to obtain a smoother

and more stable explanation. Then $x_i$'s importance score is calculated as follows (Smilkov et al., 2017):

$$I(x_i) = \frac{1}{n} \sum_1^n \frac{\partial f(X + \mathcal{N}(0, \sigma^2))}{\partial x_i} \quad (2)$$

- **Locally Interpretable Model-Agnostic Explainer (LIME)** revolves around using an explainable model, such as a linear model or decision tree model, to locally approximate the prediction of the targeted black box model. This method does not delve deeply into the model. Instead, it focuses on identifying changes in the output of the black box model resulting from minor perturbations to the input and trains an explainable model at the point of interest (the original input) based on these changes. Specifically, for the input $X$, LIME samples in its neighborhood (which is considered as a perturbation) and subsequently feeds the perturbed data $X'$ to the model to obtain predictions $f(X')$. Then a simple interpretable model is trained depending on $X'$ as inputs, the predictions as labels, and the similarity between $X$ and $X'$ as weights. Finally, the parameters of the trained interpretable model $g$ can be used to explain the importance of each feature in $X$. It is represented mathematically as follows (Ribeiro et al., 2016):

$$I(X) = \arg \min_{g \in G} \mathcal{L}(f, g, \pi_x) + \Omega(g) \quad (3)$$

where $\Omega(g)$ represents the model complexity of the explainable model $g$, and $G$ denotes all possible explainable models, and $\pi_x$ defines the neighbourhood of $x$. $\mathcal{L}$ is the loss function, measuring the difference between $f(x_i')$ and $g(x_i')$.

**Non-differentiable analysis**: This optimization problem is a typical convex optimization problem that can be solved by traditional optimization algorithms such as gradient descent. However, it is worth noting that the optimization procedure is performed on a perturbed dataset and does not reflect the differential structure of the model in the entire input space. Thus we do not usually consider LIME to be differentiable itself. The reason lies in the fact that the whole optimization process contains steps such as sampling from the neighborhood of $X$, nonparametric distance measures, and so on, which cannot directly calculate differential. As a result, LIME cannot give a definite gradient value, leading to the inability to use gradient-based adversarial example generation methods for attacks.

- **SHapley Additive exPlanation (SHAP)** works primarily by exhaustively enumerating different sets of features and calculating the change in the predicted value after each game. This allows the Shapley value of any particular feature associated with a particular prediction to be defined as the average of the difference between the predictions of all possible games in which that feature is and is not included. In detail, the shapley value of $x_i$ is calculated by the following equation:

$$I(x_i) = \sum_{\boldsymbol{S} \subset \Delta \backslash \{x_i\}} \frac{|\boldsymbol{S}|!(D - |\boldsymbol{S}| - 1)!}{D!} [v(\boldsymbol{S} \cup i) - f(\boldsymbol{S})] \quad (4)$$

where $\Delta$ is the set of all features, $\boldsymbol{S}$ is a subset of the features used in the model, and $f(\boldsymbol{S})$ refers to the output value of the model under the combination of features (Lundberg and Lee, 2017).

**Non-differentiable analysis**: From this definition we cannot directly derive a gradient with respect to the input $X$, since the formula includes the traversal of all possible subsets of features, as well as the difference of the predicted values. On the other hand, each term in the SHAP formula may have different specific values due to its dependence on the feature subset $S$. Also, the SHAP formula is not guaranteed to be continuously differentiable due to the nonlinearity of the prediction model. Therefore, we can argue that the SHAP method, like LIME, is not differentiable in the traditional sense due to the design of its process which is independent of the computed gradient.

## 3.2 ADVERSARIAL ATTACK ON EXPLANATION

The goal of adversarial attacks in AI is to add small perturbations or noise to the input data that causes a change in the output of the model. Correspondingly, in XAI, the goal of an adversarial

attack is to add small perturbations to the data so that the explanation changes while the output of the black-box model remains the same (Ghorbani et al., 2019).

In this paper, we consider regression tasks, in particular, on time series. For input data $x \in \mathbb{R}^T$, a trained black-box model $f$ will predict the value of $x^{T+1}$. The post-hoc XAI will then generate the importance $I$ of each time point of $x$ according to $f$. Then the adversarial attack on the explanation can be described as follows:

$$\arg \max_{x'} \mathcal{D}(I(x, f), I(x', f))$$
$$s.t. \quad \| f(x) - f(x') \| < \delta, \quad \| x - x' \|_\infty < \epsilon \tag{5}$$

where $x'$ is the adversarial example generated by the adversarial attack, and $\mathcal{D}$ denotes the distance between the original and the adversarial explanation. The first constraint is to make the model's outputs for $x$ and $x'$ approximate, as a way to ensure that the attack is not easily detectable. This constraint is usually $f(x) = f(x')$ in classification tasks, but in regression tasks where $f(x)$ is usually continuous, making the model output completely invariant is a stringent condition, so here it is sufficient to make the change in output less than a threshold $\delta$. Another constraint is intended to limit the size of the perturbation, thus generating changes that are not easily detectable.

### 3.3 EXPLANATIONS DISCREPANCY MEASUREMENT

To quantify the difference between pre and post perturbation explanations, there are two commonly used measurements. These measurements can be applied as the optimization objective function in Eq.5, known as $\mathcal{D}$.

- **Top-$k$** considers the most important $k$ elements of the explanation as a set. It calculate the discrepancy between the two explanations by

$$\mathcal{D}_{top-k}(I(x), I(x')) = -\frac{I(x) \cap I(x')}{k} \tag{6}$$

Using top-$k$ as a metric reduces the importance score corresponding to the most important elements. The range of this metric is $[0, 1]$, with smaller values representing a greater difference between the two explanations.

- **Center of Mass** treats the explanation as a diagram. The center of mass of a $W * H$ explanation is defined as $\mathcal{C}(I(x)) = \sum_{i \in \{1,...,W\}} \sum_{j \in \{1,...,H\}} I(x)_{ij}[i, j]^T$, and the discrepancy between explanations is (Ghorbani et al., 2019):

$$\mathcal{D}_{com}(I(x), I(x')) = \| \mathcal{C}(I(x)) - \mathcal{C}(I(x')) \|_2 \tag{7}$$

The center of mass attack causes the explanation to deviate as much as possible from the original center of mass, and this attack is mainly used for image data. A larger distance between two centers of mass represents a larger difference in explanations.

## 4 PSO BASED LOCALLY ATTACK ON TIME SERIES

Our primary goal is to design an adversarial example generation method that does not require gradient computation. To achieve this, we consider employing an evolutionary algorithm based on a search strategy to solve the optimization problem in Eq.5. XAI methods, particularly those relying on perturbations, are often time-consuming and computationally expensive. In generating adversarial examples, it is necessary to repeatedly obtain explanations for the perturbed samples, significantly increasing the computational burden. Thus, selecting a more efficient optimization algorithm is crucial. Among various evolutionary algorithms, PSO offers several advantages over alternatives such as genetic algorithms and differential evolutionary algorithms. It has fewer parameters, reducing the time spent on parameter tuning. Additionally, PSO strikes a better balance between global and local search while converging faster than genetic algorithms. Moreover, PSO's particle update mechanism is simpler to parallelize, which can further reduce computational time. For these reasons, we ultimately select PSO (Kennedy and Eberhart, 1995) to address this optimization problem.

PSO involves five key steps: initialization, evaluating fitness, updating individual and global optima, updating particle velocity and position, and checking termination criteria. The middle three steps

are repeated iteratively until the termination conditions are met. For further details, please refer to Alg.1

**Initialization:** In this step, we initialize the particle swarm's position, velocity, inertia factor, acceleration constants, and the maximum number of iterations. While traditional PSO typically initializes particle positions randomly within the search space, we modify this by setting the initial position of each particle to match the original sample. This approach increases the likelihood of generating adversarial examples that closely resemble the original sample. Additionally, we implement a local attack strategy to make the attack less detectable in time series data. Instead of perturbing all time points, we select only a certain percentage of time points to perturb. These time points are chosen based on their significance in the XAI-generated explanation, focusing on reducing the importance of these critical time points.

**Evaluating fitness:** The core of the PSO algorithm lies in the design of the fitness function, which directs the search path for the particles. The fitness function typically measures the difference between the original and adversarial explanations, with particles searching for solutions that maximize this difference. According to Eq.5, the fitness function is generally designed as the difference between two explanations, and the particle will search in the direction where the difference is greater. In Sec.3.3, we discuss two commonly used fitness functions, top-$k$ and center of mass, but they may not be ideal for time series data. For time series data, XAI explanations indicate the importance of individual time points, which can be interpreted as a time series themselves. Thus, we employ Dynamic Time Warping (DTW) (Berndt and Clifford, 1994), a method for measuring the similarity between time series, to evaluate the difference between the original and adversarial explanations.

Dynamic Time Warping (DTW) aims to find an optimal alignment between two sequences by minimizing the distance between corresponding elements after alignment. For the original sample's explanation $I(x)$ and the adversarial example's explanation $I(x')$, we assume that both sequences have a length of $T$. To capture the differences between the two explanations, we define a cost matrix $M \in R^{T \times T}$, where each element $M[i, j]$ represents the cumulative distance between the first $i$ elements of $I(x)$ and the first $j$ elements of $I(x')$. The value of the $M[i, j]$ is computed iteratively, considering both the current difference and the optimal path from the preceding elements. The computation of $M[i, j]$ can be formalized as follows:

$$M[i, j] = \min(M[i-1, j-1], M[i-1, j], M[i, j-1]) \\ + d(I(x_i), I(x'_j)) \tag{8}$$

where $d(\cdot)$ denotes the Euclidean distance between two elements. If the sample is a time series of $F$-dimensional variables, we can compute a DTW matrix for each variable, denoted as $M_f$. Then the difference between the two explanations can then be expressed as:

$$\mathcal{D}_{dtw}(I(x), I(x')) = \frac{1}{F} \sum_{f=1}^{F} M_f[T, T] \tag{9}$$

And we use this as a fitness function for PSO. It is worth noting that the constraint included in Eq.5 requires $\| f(x) - f(x') \| < \delta$. We designed this into the fitness function, and if the particle cannot satisfy this constraint, we set its fitness value to a meaningless minimum and do not continue to compute the DTW matrix.

**Updating the individual optimum and the global optimum**: For each particle, the current fitness value is compared to its historical best fitness value. If the current fitness is better, the particle's individual best is updated. Similarly, the global best across all particles is also updated if any particle's individual best surpasses the previous global best.

**Update the velocity and position of the particle**: Knowing the individual optimum and global optimum of the particle, the particle position is updated using the inertia weights $w$, the individual learning factor$c_1$, and the global learning factor$c_2$. Note that during the update, the position of the particles cannot be beyond the search range. We limit the size of the search space using the constraint $\| x - x' \|_\infty < \epsilon$ in Eq.5.

**Termination**: PSO will terminate when the number of iterations is exhausted.

---

**Algorithm 1** PSO based locally attack on time series

---

**Input:** test time series $\mathbf{x}$, maximum norm of perturbation $\epsilon$, black box model $f(\cdot)$, XAI method $I(\cdot)$, number of iterations $N$, number of particles $P$, inertia weights $w$, individual learning factor $c_1$, global learning factor $c_2$;

**Output:** adversarial example $\mathbf{x}'$

1: Select the most important time points $\mathbf{x}_p$ to be perturbed according to $I(x)$, and the remaining points are in $\mathbf{x}_r$.
2: For every particles $i = \{1, \ldots, P\}$, initialize positions $\mathbf{pos}[i] = \mathbf{x}_p$ and speed $\mathbf{spd}$.
3: Set particle best location **pbest** and fitness **pbest_val**.
4: Set global best location **gbest** and fitness **gbest_val**.
5: **for** $n = 1 \rightarrow N$ **do**
6:    Set particle rotation $\mathbf{r}_1 = rand()$, $\mathbf{r}_2 = rand()$.
7:    $\mathbf{spd} = w * \mathbf{spd} + c_1 * \mathbf{r}_1 * (\mathbf{pbest} - \mathbf{pos}) + c_2 * \mathbf{r}_2 * (\mathbf{gbest} - \mathbf{pos})$.
8:    $\mathbf{pos} = \text{Clip}(\mathbf{pos} + \mathbf{spd})$.
9:    Fuse the **pos** with $\mathbf{x}_r$ in the temporal order of $\mathbf{x}$ to obtain an adversarial example $\mathbf{x}_n$.
10:    Get $\mathcal{D}_{dtw}(I(\mathbf{x}), I(\mathbf{x}_n))$.
11:    Compare and update **pbest**, **pbest_val**, **gbest**, **gbest_val**.
12: **end for**
13: Fuse the **gbest** with $\mathbf{x}_r$ in the temporal order of $\mathbf{x}$ to obtain the final adversarial example $\mathbf{x}'$.
14: **return** $\mathbf{x}'$

---

## 5 EXPERIMENTS AND RESULTS

### 5.1 EXPERIMENT SETUP

In our experiments, we selected two multivariate time series datasets for prediction. The first dataset consists of a multivariate time series of PM2.5 levels in Beijing[1]. This dataset includes 43,824 records across 12 variables, collected from January 1, 2010, to December 31, 2014, at an hourly sampling rate. The second dataset contains tick data of stock prices from the Shenzhen stock market, collected in July 2022. We selected stock code SZ000001 for stock price prediction and explanation. The dataset spans from July 1, 2022, to July 31, 2022, covering 21 trading days (excluding weekends). The data is collected approximately every 10 seconds, with 5 variables recorded.

Next, we employed three black-box models for time series classification: Transformer, TCN, and LSTM with input cell attention. Four XAI methods, mentioned in Sec.3.1, were used to generate the explanations.

We then applied the proposed adversarial attack method to these explanations. To compare the effectiveness of using DTW as the objective function, we also tested the top-$k$ and center of mass functions. For all experiments, the number of iterations was set to 200, with $\delta$ and $\epsilon$ set to 0.0005 and 0.1, respectively.

### 5.2 EVALUATION METRICS

In this paper, we evaluate the effectiveness of the attack methods and the robustness of the explanations using explanation discrepancy metrics. Explanation discrepancy metrics are the current mainstream approach to assessing explanation robustness. The most commonly used metrics include top-$k$ intersection, Spearman's Rank-Order Correlation, and L2 distance. The top-$k$ intersection metric is equivalent to Eq.6. This metric evaluates how many of the top $k$ important features are shared between the original and adversarial explanations. Spearman's Rank-Order Correlation measures the strength of the relationship between two ranking variables. For this metric, the explanations $I(x)$ and $I(x')$ are converted into the ranking vectors $R(x)$ and $R(x')$, and the correlation between the two vectors is calculated using Spearman's correlation coefficient. Both the top-$k$ intersection and Spearman's correlation metrics have a range of [0,1], where smaller values indicate a more successful attack. Finally, L2 distance is a direct measure of the normalized distance between the two explanations, with larger values indicating a more successful attack. These metrics provide a com-

---

[1]https://archive.ics.uci.edu/dataset/381/beijing+pm2+5+data

Table 1: Robustness of explanations generated by different combinations of models and XAI under DTW attack objective functions

| Model | XAI | PM 2.5 Data | | | SZtick Data | | |
|---|---|---|---|---|---|---|---|
| | | TKI | SRC | L2 | TKI | SRC | L2 |
| LSTM | LIME | 0.6761 | 0.8125 | 3.761 | 0.6708 | 0.6627 | 3.795 |
| | SHAP | 0.2079 | 0.0102 | 3.695 | 0.3147 | 0.0170 | 3.532 |
| | SM | 0.8800 | 0.6630 | **0.486** | 0.8783 | 0.6428 | 1.352 |
| | SG | **0.9658** | **0.9975** | 0.690 | **0.9700** | **0.9963** | **1.341** |
| TCN | LIME | 0.6808 | 0.8157 | 3.868 | 0.6744 | 0.6746 | 3.745 |
| | SHAP | *0.1981* | *-0.0056* | *3.907* | *0.2972* | *0.0027* | 3.852 |
| | SM | 0.5092 | 0.3577 | 3.194 | 0.3807 | 0.2039 | *4.904* |
| | SG | 0.5425 | 0.3842 | 3.221 | 0.6679 | 0.7111 | 3.560 |
| Transformer | LIME | 0.6714 | 0.8144 | 3.711 | 0.6838 | 0.6906 | 2.076 |
| | SHAP | 0.2086 | 0.0064 | 3.826 | 0.3103 | 0.0077 | 3.484 |
| | SM | 0.4793 | 0.3489 | 2.149 | 0.4629 | 0.2451 | 3.127 |
| | SG | 0.6101 | 0.6101 | 2.071 | 0.4660 | 0.2778 | 3.109 |

prehensive way to evaluate both the success of the attack and the robustness of the XAI-generated explanations.

## 5.3 RESULTS ON DIFFERENT MODELS AND XAI METHODS

We begin by attacking the explanations generated by the combination of the three models and the four XAI methods across two datasets using the $\mathcal{D}_{dtw}$ metric. The robustness of the explanations is assessed and summarized in Table 1. The attack strategy perturbs the 20% most important time points as determined by the explanations. In the table, the explanations with the highest robustness are highlighted in bold, while the least robust explanations are italicized.

It is evident that LSTM with SG generates the most robust explanations. The Top-K Intersection (TKI) and Spearman's Rank-Order Correlation (SRC) metrics close to 1 indicate that the explanation remains almost unchanged after slight perturbations to the input sample. This robustness can be attributed to SG, which reduces the uncertainty in gradient explanations by calculating multiple gradients and averaging them. Additionally, LSTM's gating mechanism and memory cell effectively prevent the gradient vanishing and exploding problems, contributing to more stable explanations. Consequently, gradient-based explanations like SM and SG are more robust when paired with LSTM. In contrast, TCN and Transformer models are less effective at handling long time sequences. The convolutional kernel in TCN and the attention mechanism in Transformer can introduce sparsity in information, which hinders gradient propagation through the deep network and reduces the stability of SM and SG explanations.

On the other hand, the perturbation-based methods, LIME and SHAP, are model-agnostic and focus on feature interactions and importance. As a result, their explanations show similar robustness across the three AI models. However, the results also demonstrate that both LIME and SHAP explanations are relatively unstable, underscoring the effectiveness of our proposed attack. Specifically, SHAP-generated explanations diverge significantly from the original explanations when facing adversarial samples. Overall, our attack is successful not only against non-differentiable methods like LIME and SHAP but also against gradient-based methods (except for LSTM+SG). This highlights the robustness weaknesses in existing XAI techniques and offers insights for improving future XAI methods.

## 5.4 RESULTS ON DIFFERENT OBJECT FUNCTIONS

The robustness of the explanations obtained with three different functions as objectives against the attack is shown in Table 2. In this experiment, which focuses on assessing the effectiveness of the attack, we highlight the explanations with the worst robustness in bold. The results particularly

Table 2: Robustness of explanations generated by LSTM and two XAI methods under different attack objective functions

| XAI | Obj. Func. | PM 2.5 Data | | | SZtick Data | | |
|---|---|---|---|---|---|---|---|
| | | TKI | SRC | L2 | TKI | SRC | L2 |
| LIME | DTW | **0.6761** | 0.8125 | 3.761 | **0.6708** | **0.6627** | **3.795** |
| | TopK | 0.6809 | **0.8122** | **4.056** | 0.6783 | 0.6719 | 3.770 |
| | COM | 0.6789 | 0.8208 | 3.840 | 0.6775 | 0.6630 | 3.615 |
| SHAP | DTW | 0.2079 | 0.0102 | 3.695 | 0.3147 | 0.0170 | 3.532 |
| | TopK | 0.2118 | 0.0080 | 3.595 | 0.3175 | 0.1293 | 3.416 |
| | COM | 0.2178 | 0.0103 | 3.636 | 0.3197 | 0.0185 | 3.402 |

emphasize the explanations produced by LIME and SHAP when applied to the LSTM model. The attack strategy targets the most important 20% of time points in the dataset. While the differences resulting from the use of different attack objective functions are minimal, we can observe that the $\mathcal{D}_{com}$ function yields the poorest performance. This outcome is primarily due to the fact that this objective function is better suited for image data rather than time series data. Furthermore, the local attack strategy aligns with the core principle of the $\mathcal{D}top - k$ function, which tends to favor the $\mathcal{D}top - k$ attack. As a result, the findings are very similar to those produced by the $\mathcal{D}_{dtw}$ objective function.

## 5.5 Results on Global Attack and Local Attack

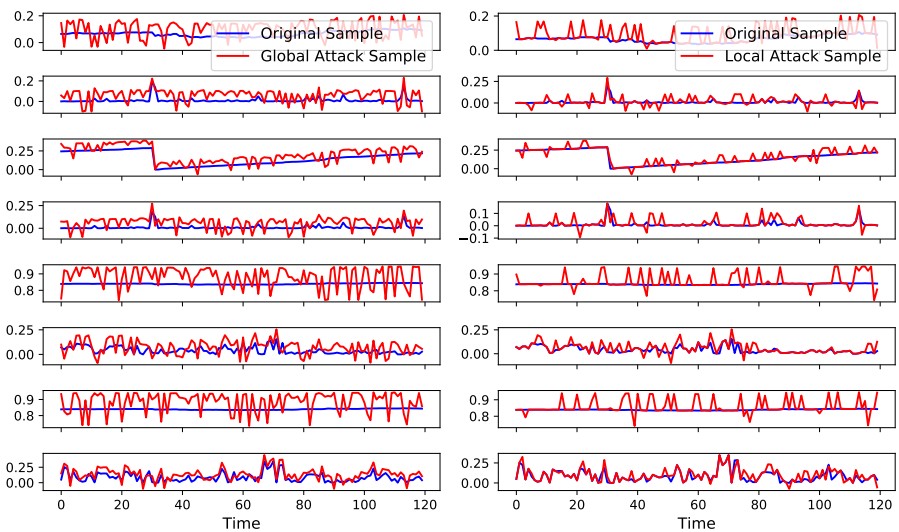

Figure 2: This figure shows comparison of original sample, local attack adversarial sample and global attack adversarial sample.

To highlight the advantages of the local attack strategy, we also compare it with the global attack strategy. The experiment targets the LSTM model with the KS explanation and attacks with DTW. Fig. 2 shows the adversarial samples of a particular original sample in the SZtick dataset under local time point and global time point adversarial attacks. The blue curve in the figure represents the original sample and the red curve represents the global attack sample and local attack sample, respectively. We visualize the difference between the explanations generated by the original sample and the adversarial sample in the form of a heat map, as shown in Fig. 3. The color shade of the heat map indicates the level of difference in importance at that point in time, with darker colors indicating larger differences.

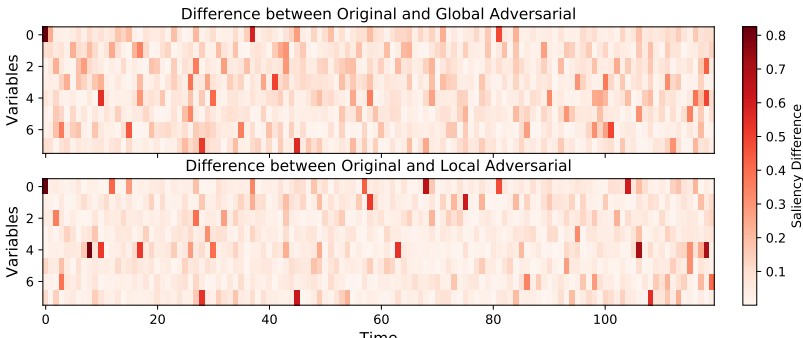

Figure 3: Heatmap of the difference between original and adversarial explanations under different attack strategies.

It can be observed from the figure that the distribution of differences between global and local attacks across time points and variables is overall similar. Although global attacks produce larger differences at more points in time, local attack can achieve comparable results at specific points in time. This suggests that local attack are comparable to global attack at certain key points, thus achieving significant interference with the original sample-generated explanations.

With similar attack effects, it can be seen from Fig. 2 that the local attack exhibits larger fluctuations at specific time points compared to the global attack, but the overall trend remains stable. This indicates that the local attack is able to maintain the overall structure of the time series when interfering with a specific target, thus reducing the impact on less important time points. In addition, the computational complexity and time cost of the local attack strategy are significantly lower compared to the global attack, which makes the local attack more practical in resource-constrained situations. In terms of stealth, local attacks can achieve the attack purpose without significantly changing the overall data distribution, which is more difficult to be detected, making it more threatening in practical applications.

## 6 CONCLUSIONS

In this paper, we propose a gradient-free adversarial attack method targeting explanations generated by explainable artificial intelligence (XAI). We experimentally demonstrate that our attack can significantly alter the explanations produced by XAI methods with only minor modifications to the input samples. This approach not only enhances the stealthiness of the attack but also improves its efficiency and applicability. Our experiments reveal the non-robustness of time series explanations, highlighting vulnerabilities that can be exploited. These findings lay the groundwork for future research aimed at developing defenses against such adversarial attacks on time series explanations.

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
