# OpenReview forum: "Gradient-Free Adversarial Attack on Time Series Regression: Targeting XAI Explanations"
_ICLR.cc/2025/Conference — ICLR 2025 Conference Withdrawn Submission_

### Official Review · Reviewer_gWSG · 2024-10-30

**Soundness:** 4
**Presentation:** 1
**Contribution:** 4
**Rating:** 6
**Confidence:** 5

**Summary:**

This paper introduces a black-box attack to manipulate the output of explanation methods for time series regression. It relates to previous work on crafting adversarial examples for explanations of image classification using gradient-based optimization methods. Both the solution of using PSO, and the setting of time series regression, are novel in this line of work. Extensive experiments with 3 models (LSTM, TCN, Transformer) and 4 XAI methods show that popular algorithms are vulnerable to adversarial examples, which undermines their applicability, and facilitates future work on robust explanations in time series.

**Strengths:**

1. The idea of using PSO to optimize for attacking explanations is interesting. Usually, in related papers, unrealistic assumptions are made about the white-box access to the model's weights.
2. Focusing on XAI for time series regression is very original.
3. It is commendable that the experiments already span four diverse explanation methods (LIME, SHAP, Saliency, SmoothGrad) and three model families (LSTM, TCN, Transformer), which show valuable comparisons.
4. The paper is easy to read; figures and tables are appropriate.

**Weaknesses:**

1. **Experiments.** PSO is a random algorithm. How many random repetitions were initiated in experiments? Are metric values reported in Tables 1 & 2 aggregated means? Please add standard deviations to the analysis.
2. **Code.** To the best of my knowledge, the paper is not supplemented with code that implements the method and allows to reproduce the experimental results. Can you share the code, e.g. on Anonymous GitHub?
3. Overall, the **presentation** is poor (see suggestions below), and I count on it being fixed not to obfuscate the valuable contribution.

**Questions:**

I am willing to increase my score if the paper's presentation is significantly improved.

1. Rephrase "XAI explanations" (6 times in the paper), which sounds like a pleonasm. "XAI methods", "XAI techniques" make more sense.
2. Fix Eq.(4). where $v()$ is not defined and $f(S)$ makes no sense. I suggest to define $v()$ using $f()$ and use this $v()$ in Eq.(4).
3. I am confused about the use of $X$ and $x$. L182: Do you mean to write $\pi_X$ instead? L219: Here $x$ is introduced, but $X$ was used in the previous section; please unify the notation.
4. How is $I(x) \cap I(x')$ defined? Why is there a minus sign, but the metric's range is [0, 1]?
5. L304: use a different letter than $f$ to denote $M_f$, which was earlier used to denote the model function $f$.
6. Report model predictive performance results on training and test sets (3 models x 2 datasets).
7. L479: what is the "KS explanation"? Do you mean "SHAP"?
8. Where is the "global attack" (evaluated in Sec. 5.5) described exactly? Please clarify it in Section 4.
9. Please define a threat model under which the attacker operates. For example, what can be accessed by an attacker: an input sample, a dataset, a neural network model? See [1-5] for a few examples of discussing such a threat model in different papers on adversarial ML:
    [1] Glaze: Protecting artists from style mimicry by text-to-image models
    [2] Extracting training data from diffusion models
    [3] Extracting training data from large language models
    [4] RAB: Provable robustness against backdoor attacks
    [5] Local model poisoning attacks to byzantine-robust federated learning, etc.

### Other feedback
- L49: typo in "unchanged.(Ghorbani et al., 2019)."
- L53: missing space in "method(Huang et al., 2023)"
- L80: typo in "Fig.??,"
- L106: missing full stop between " Sec.3 Sec.4"
- "Locally Interpretable Model-Agnostic Explainer" should be "Local Interpretable Model-agnostic Explanations"
- L172: missing space in in "X,LIME"
- L221: missing comma in "Then, the adversarial"
- Eq.(5) clarify that you write $I(x, f)$ to emphasize explaining model f. Instead, you could also write $I_f(x)$ or $I(x; f)$
- The title of Section 3.3 is capitalized, while the titles of Sections 3.1 & 3.2 were not.
- L239: missing "s" in "It calculate the"
- Eq.(6) write "\mathcal{D}_{\mathrm{top-k}}" instead (also in L451 etc.)
- The title of Section 4 is not capitalized, while the titles of Sections 2 & 5 are capitalized.
- L320: missing spaces in "factorc1" and "factorc2."
- Typo in the title of Section 5.4. – do you mean objective functions?

---

### Official Review · Reviewer_gvhW · 2024-11-01

**Soundness:** 2
**Presentation:** 2
**Contribution:** 2
**Rating:** 3
**Confidence:** 3

**Summary:**

This paper proposes a novel gradient-free adversarial attack method to test the robustness of Explainable AI (XAI) explanations for time series regression problems. The proposed method uses Particle Swarm Optimization to generate adversarial examples without needing gradient information, making it more effective for real-world scenarios and non-differentiable XAI techniques.

**Strengths:**

- it is interesting to use PSO to solve XAI problem.

**Weaknesses:**

- limited novelty. This paper only uses DTW as the distance for PSO.
- limited experiments. The baselines and datasets for Figure 1 and Figure 2 are not enough.
- why do the authors use DTW as the distance between two time series? could authors provide a theoretical analysis about what properties of DTW make it optimal compared with other distances, such as MAE (RMSE), cosine similarity?
- The authors claim that people can easily detect subtle perturbation in Line 79 and provide a figure to validate it. However, in this figure, the time series is a smooth periodic function (sine function), and it is the smoothness and period that make the perturbation so obvious. In common time series, these good properties may not exist and noise would be everywhere. Could you use a time series in one real-world dataset, such as Traffic/Weather, to draw the same figure? let us see whether we can have the same conclusion then (unnoticeable in image but obvious in time series).
- typos: Figure reference is broken in line 80.

**Questions:**

The same as weakness.

---

### Official Review · Reviewer_8TPe · 2024-11-03

**Soundness:** 2
**Presentation:** 3
**Contribution:** 1
**Rating:** 3
**Confidence:** 4

**Summary:**

This paper proposes a black-box adversarial attack on Explainable Artificial Intelligence (XAI) methods for time series regression models. Previous studies on XAI attacks have primarily focused on white-box settings and models in computer vision. However, attacks on time series models in black-box settings remain largely unexplored. To address this gap, the authors adapt the Particle Swarm Optimization (PSO) black-box optimization algorithm for such attacks. Specifically, they initialize the algorithm with the original time series instead of zeros to improve local search performance. They also employ Dynamic Time Warping (DTW) as the objective function for PSO. Experimental results on several combinations of models and XAI methods demonstrate the effectiveness of the proposed method.

**Strengths:**

- **Novel research problem**: Black-box adversarial attacks against XAI methods for time series regression models have not yet been extensively studied.
- **Well-written paper**: The paper is well-organized and easy to follow.

**Weaknesses:**

- **Lack of justification for methodology design**: The choice of DTW in PSO is not explained. It appears to be selected solely because of its improved performance over top-K or center of mass approaches. See Question 1 for further details.
- **No consideration of defense mechanisms**: The authors do not discuss potential defenses that could detect or reject adversarial examples.
- **Significant adversarial perturbation**: In Figure 2, the generated adversarial examples deviate significantly from the original samples, making them potentially easy to detect with defense methods.
- **Limited theoretical or technical contribution**: Given the weaknesses noted above, the paper’s contribution to attacking XAI methods appears limited in terms of theoretical or technical advancements. Overall, it reads more as an application of black-box adversarial attacks on XAI methods for time series regression models.

**Questions:**

- The reasoning behind using DTW as the fitness function is missing. Given that differences between two time series explanations should ideally be compared step-by-step, the cumulative distance measured by DTW may not align well with the objective of perturbing XAI methods effectively. A more in-depth analysis on the rationale for incorporating DTW would be beneficial.
- Minor typos:
  - In line 80-81, the figure reference is missing.

---

### Official Review · Reviewer_Qib6 · 2024-11-05

**Soundness:** 3
**Presentation:** 2
**Contribution:** 2
**Rating:** 3
**Confidence:** 3

**Summary:**

The paper introduces a gradient-free adversarial attack method designed to target non-differentiable XAI techniques in time series regression problems. The author propose a novel gradient-free adversarial attack method specifically designed for time series explanations, targeting non-differentiable XAI techniques. The paper also introduces a Dynamic Time Warping (DTW) based objective function and a local attack strategy to enhance the effectiveness of the attack on time series data. The experiments conducted across three black-box models and two time series datasets demonstrate the vulnerability of current non-differentiable XAI methods and show the superiority of the proposed approach over existing attack methods.

**Strengths:**

This paper is trying to solve a critical question in the XAI robustness domain, which is well-motivated.

**Weaknesses:**

The paper structure is poor and the not well-organized. Too much pages are used on related work and preliminary. The paper writing is not standard.
The methods seems to be lack of novelty. PSO is an existing method for black-box attack. The proposed method uses DTW of explainable result of X and X_adv as loss function.
The whole experiment setup is not very clear. There is no baseline comparison. No results to support the effectiveness of proposed methods.  Table1 evaluated the robustness of different XAI models under DTW attack objective, but this is not what you what to show. What you want to show in this paper is the effectiveness of your method compared to other attack methods.  Table 2 compared different objectives, but still cannot show the effectiveness of DTW loss.  Your experiments cannot support your claims in the contribution.

The author employed three black-box models for time series classification: Transformer, TCN, and LSTM with input cell attention. However, these models are not the SoTA method for time series classification, the author may focus on more advanced models.

**Questions:**

As shown in the weakness section.

---

### Note · Authors · 2024-11-21

I have read and agree with the venue's withdrawal policy on behalf of myself and my co-authors.